# Experimental Study on Phase Change Material with Solar Heater System for Building Heating

**Haijun Han** [1,2], **Hongyan Zhou** [2], **Ouyang Dong** [2] and **Junjie Ma** [1,*]

1  College of Science, Hainan Tropical Ocean University, Sanya 572000, China
2  Qinghai Institute of Salt Lakes, Chinese Academy of Sciences, Xining 810008, China
*  Correspondence: junjiema1986@126.com; Tel.: +86-18720778081

**Abstract:** An integrated solar heating system with a new type of phase change material (PCM), solar collectors and test building were developed. The exothermal and endothermal behaviors of the PCM were determined, and the stability and comfort of the solar heating system were researched. The integrated solar heating system was operated on the test building heating for one heating period, and the temperature of heating rooms, the outdoors, and the contrast rooms were recorded and collected by a data acquisition system. The collected temperature data indicated that the integrated solar heating system with PCM could produce heating stability and continuity; the average temperature of the heating rooms using PCM was 4.6 °C higher than the contrast rooms, which did not use PCM. Taking 16 °C as the lowest standard room temperature, the integrated solar heating system could save approximately 45% of energy during one heating period. The successful development of an integrated solar heating system, coupled with phase change materials and solar collectors for building heating will lay a solid foundation for achieving the goals of building energy conservation and "carbon peaking and carbon neutrality".

**Keywords:** phase change material; building heating; solar energy; heating season





## 1. Introduction

Energy is an important foundation for social progress and economic development. At present, the world energy consumption structure is undergoing major changes. With the rapid increase of the proportion of photovoltaic and wind power generation in the energy structure, major countries in the world have put forward carbon emission targets to develop "carbon peaking and carbon neutrality" [1]. More importantly, the proportion of renewable energy targets worldwide has been continuously raised and advanced. However, renewable energy, such as photovoltaic, wind power and other new energy sources, is characterized by volatility and instability. Energy generation and energy utilization do not match in time and space, and energy storage can better solve this problem. Among various energy storage technologies, heat storage, especially phase change heat storage technology, as an effective method of energy utilization, has attracted more and more attention. It is widely used in many fields, such as clean heating, energy saving in buildings, waste heat utilization, temperature control, and others.

Xining City in Qinghai Province, located in the eastern part of the Qinghai-Tibet Plateau in China is a severely cold area, and the heating time in winter is up to six months. Energy consumption of heating in winter is the main part of the total energy consumption of buildings. At the same time, Xining City is a zone with extremely abundant solar energy resources, and annual solar total radiation is 5836.3 MJ/m²·a [2]. Therefore, using solar energy to supplement energy for building heating in winter is one of the most important measures for energy saving and emission reduction in buildings. However, there are some difficulties and disadvantage in the utilization of solar energy. Firstly, as the solar energy reaching the ground is restricted by natural conditions such as day and night, seasons,

geographical latitude and altitude, as well as the random influence of cloud and rain, the solar radiation reaching a certain ground is both discontinuous and extremely unstable. Secondly, the total solar radiation reaching the earth's surface is large, but the energy flow density is very low. Thirdly, at present, the development level of solar energy utilization is theoretically feasible and technically mature, but some energy utilization devices are inefficient because of their high cost. Therefore, a better use of solar energy is necessary to solve the intermittent and unreliable solar energy supply. Phase change energy storage materials can automatically absorb or release latent heat into the environment, so they can alleviate the contradiction between energy storage and energy consumption in time and space, and can achieve the purpose of improving the indoor thermal environment and saving building materials.

In recent years, researchers [3–9] have carried out a lot of active and effective study work on how to combine phase change heat storage technology with solar energy thermal utilization, and turn the unstable solar energy into a stable heating source. In the 1950s, the United States, Japan, and other developed countries began to study the combination systems of solar energy and phase change energy storage materials [3]. Youssef et al. [4] reported an indirect solar heating system with a phase change material heat exchange tank. The research found that the heating performance and operation stability of the system had been improved. Wu et al. [5] reported the effects of different heat storage materials on the performance of solar heat pump systems. Fu et al. [6] reported a phase change energy storage solar heat pump system. It was found that the phase change energy storage box could play a role of peak shaving and valley filling for solar energy utilization, and improve the operational stability of the system. Kuznik et al. [7] reported the thermal environment of buildings with phase change materials and ordinary buildings. The results showed that phase change materials could not only improve the indoor natural convection heat transfer, but also improve the quality of the indoor thermal environment. Kapsalis et al. [8] reported that PCM could improve the operational stability of heat pumps and reduce energy consumption. The development of PCM assisted Evacuated tube solar dryer (ETSD). Malakar et al. [9] reported the development and application of PCM assisted evacuated tube solar dryers. In this work, based on the previous research on the thermal performance of PCM at room temperature, the integrated solar heating system coupled with a solar water heater, PCM and building heating system was developed. A new type of inorganic salt PCM with good storage and long cycle life was used for testing the integrated solar heating system. The system was tested for more than six months in the heating period.

## 2. Experimental System Design (Experimental Setup and Data Analysis Procedure)

The effectiveness of PCM was examined in the experimental platform (Figure 1) coupling of heating buildings test rooms, phase change materials and solar collectors. As shown as Figure 1, the rooms from left to right are A, B, C, D and E: room A has no PCM installed and is used as the contrast; room B is an equipment room fitted with a computer and temperature collectors; rooms C, D and E are equipped with radiators containing PCM and connected to the solar collectors. The tests were driven by three aspects. Firstly, the basic approach of the experiment was to examine the endothermic and exothermic effectiveness of PCM. Secondly, the stabilization effect of the experiment to examine the effectiveness of PCM was evaluated by recording temperatures of the test rooms C, D and E. Finally, the energy saving effect of PCM was assessed by comparing the temperatures in the contrast and test rooms. The experimental results obtained regarding these three aspects led us to discover highly efficient practical applications of PCM. When installing an integrated solar heating system in a room, the furniture, solar radiation, amount of PCM and the number of occupants affect the temperature of test rooms directly. However, this study focused on the temperature stabilization effect and energy saving effect with and without this integrated solar heating system, so other influencing factors were not considered.

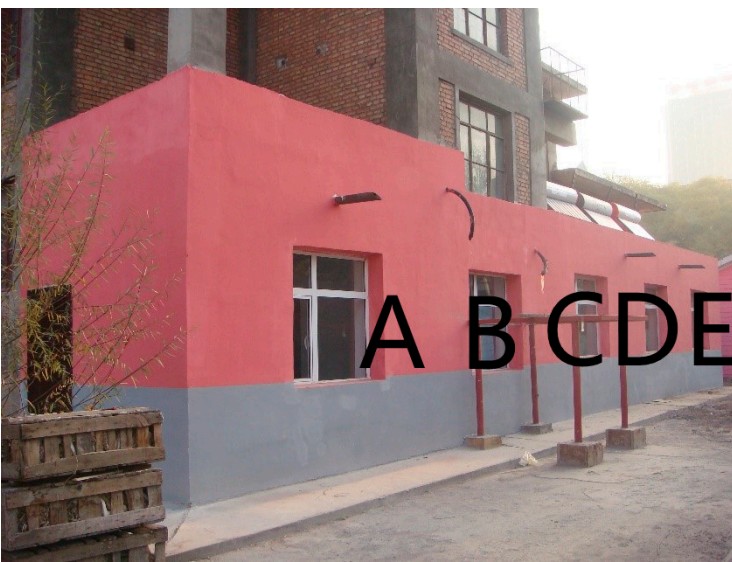

**Figure 1.** The experimental platform in this work.

### 2.1. Overview of the Experimental Facility

The experimental testing platform was constructed half a month before the heating season in Xining City, Qinghai Province, China. All of the rooms had south-facing windows and doors on their western walls and were closed after the test began. To balance the solar radiation conditions in the three huts, two walls were built, one on the west side of room A and one on the east side of room E, using the same construction materials. Every room of the experimental platform was 3000 mm × 3000 mm. The exterior walls were built of 370-mm red brick and externally-attached 50-mm extruded polystyrene (XPS). The roof was made of 200-mm reinforced concrete and externally-attached 100-mm extruded polystyrene (XPS). The interior walls between rooms were built of 240-mm brick without extruded polystyrene (XPS). Each room had a window opening to the south in the middle of the south wall. The window was made of aluminum alloy thermal insulation in a broken bridge structure. Every window size was 1500 mm × 1500 mm and was installed with double-glazing glass. Three sets of evacuated tubular solar collectors were equipped on the roof of the corresponding rooms C, D and E. To avoid the high temperature caused by the direct sunlight to the temperature sensor, all temperature sensors were suspended from the 30-cm ceiling. Room B was an equipment room fitted with a computer and temperature collector. Rooms C, D and E were the experiment testing room fitted with radiators with PCMs, circulation pumps and temperature sensors.

### 2.2. Integrated Solar Heating System

Three sets of integrated solar heating systems were equipped, corresponding to three experimental testing rooms. The integrated solar heating system mainly consisted of an evacuated tubular solar collector, phase change materials, an aluminum radiator, circulation pump and connection pipe fittings. The evacuated tubular solar collector produced by Beijing Shen'guang Solar Group (Beijing, China) was mainly composed of a water storage tank and evacuated collector tubes, reformed and equipped on the roofs of the testing rooms. PCM was carefully prepared from purchased raw chemical reagents and used after packaging. The aluminum radiator full of water and packaged PCM was self-designed and manufactured, placed at a distance of approximately 30 mm from the eastern wall of the testing room. Circulation pumps (from Shijiazhuang Pulandi Mechanical and Electrical Equipment Co., Ltd., Shijiazhuang, China; Type: PLD-1203, DC 12 V, 12 W) were used for water circulation in the water storage tanks of the solar collector and the radiators.

The data collector system consisted of a computer, data collector and multi-point temperature sensors which could automatically record the temperatures of the testing rooms,

contrast room, the outdoor environment, and the water of each radiator. Temperature data were collected and recorded every minute. The experimental time lasted from 15th October to the end of the following 15th April. The time of switch of the circulation pumps was controlled by microcomputer. The running time was 04:00 to 09:30 and 17:30 to 24:00 every day, and designed to achieve the goal as follows: (1) When the circulation pumps were opened, the hot water entered the radiator from the solar collector, while the cold water in the radiator entered the solar collector. The water temperature of the radiator rose slightly, and the PCM absorbed heat energy and took up the solid-to-liquid phase transformation. (2) When the circulation pumps were closed, the energy exchange between the solar collector and the radiator was stopped, and the room temperature depended only on the energy of the water and the PCM in the radiator. Once the water temperature in the radiator was lower than the phase change temperature of the PCM, the PCM caused the liquid–solid phase transformation, and released the heat energy. The water temperature could then be maintained near the phase change temperature, and the room temperature could be kept relatively stable. There was no other heating equipment except the radiator in the testing room, and the integrated system ran in the heating period for a long time.

### 2.3. PCMs Installation

The PCM used herein were made of an inorganic salt complex (calcium chloride hexahydrate and ammonium chloride)-based PCM, and a suitable amount of strontium chloride hexahydrate was added, to keep the stable crystallization and melting properties. The thermal properties of the PCM are shown in Figure 2. To examine the effects of PCM application and those of different PCM application areas, no PCMs were installed in rooms A and B, whereas a 50 kg PCM was installed in the radiators of rooms C, D and E.

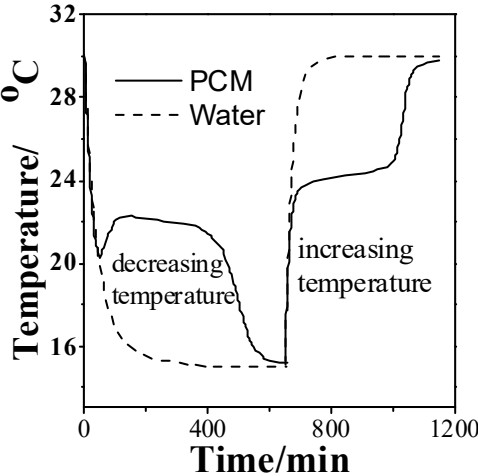

**Figure 2.** Endothermic and exothermic performance of PCM.

### 3. Experimental Results and Discussions

A continuous heating period (from 15 October to the end of the following 15 April), the experiment was performed under natural conditions. Figure 3 is diagram of the recorded temperatures of the testing rooms, contrast room and outdoor natural environment, over time. In order to facilitate observation, a short period of time was taken out for analysis, as shown in Figure 4.

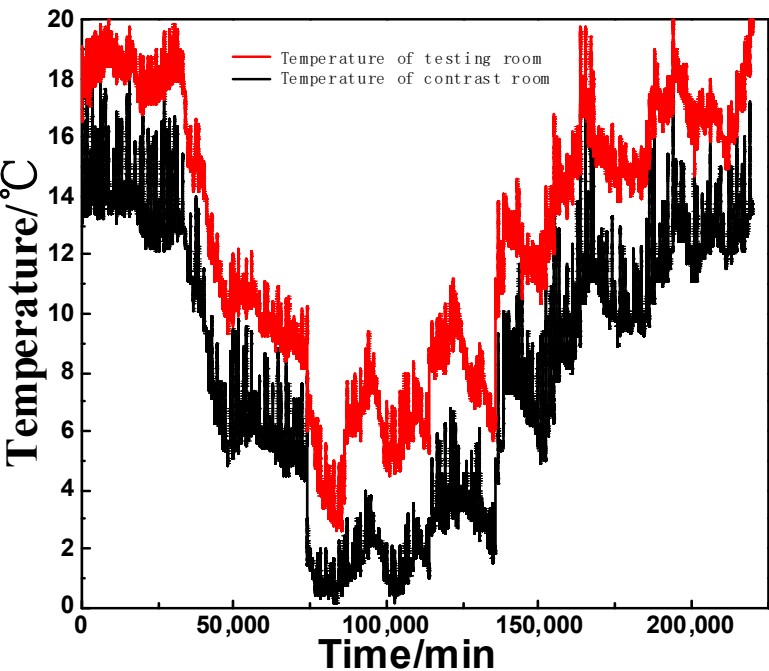

**Figure 3.** Recorded temperatures of testing rooms, contrast room and outdoor natural environment.

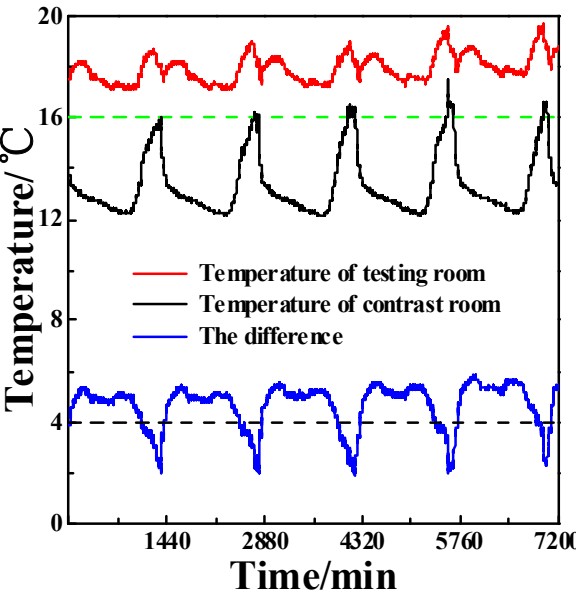

**Figure 4.** Diagram of temperatures in the testing room and contrast room over time.

It is shown in Figures 3 and 4 that the average temperature of the testing room with PCM is obviously higher than that of the contrast room temperature. The temperature fluctuation of the testing room with PCM is obviously smaller than that of the contrast room temperature. In the process of temperature rising and decreasing in the testing room with PCM, there is an approximate platform to keep the room temperature stable. This platform is caused by the endothermic and exothermic of the PCM, and the contrast room temperature is completely without this process. This indicates that the application of PCM in solar building heating can reduce room temperature fluctuations and stabilize room temperature.

### 3.1. Comparison of Temperature between Testing Room and Contrast Room

The temperature differences between the testing room and contrast room and the difference between testing room and contrast room over time is showed in Figure 4. As shown in Figure 4, the difference between the testing room and contrast room varies with time. When the room temperature rises in the daytime, the difference is small, and the minimum is about 2 °C. When the solar heating system plays a role at night and in the morning, the difference becomes larger and stays above 6 °C for a long time. When the circulation pump stops working, the room temperature is mainly heated by the PCM released energy.

### 3.2. Water Temperature in Radiator

Figure 5 shows the relation of the temperature of the testing room to the water in the radiator over time. It indicates that the temperature of the testing room with PCM rose with the rising of the water temperature. Because of the effect of the PCM, the temperature of the testing room with PCM changes slightly and the temperature changes of the testing room with PCM will lag behind the water temperature for a certain time. The phase transformation temperature of PCM is 24 °C. When the water temperature is higher than 24 °C, the PCM will absorb heat energy and take part in solid–liquid phase transformation, so that the water temperature does not rise too much. When the water temperature is below 24 °C, the PCM will undergo liquid–solid phase transformation and release heat energy, and the water temperature will remain at about 24 °C.

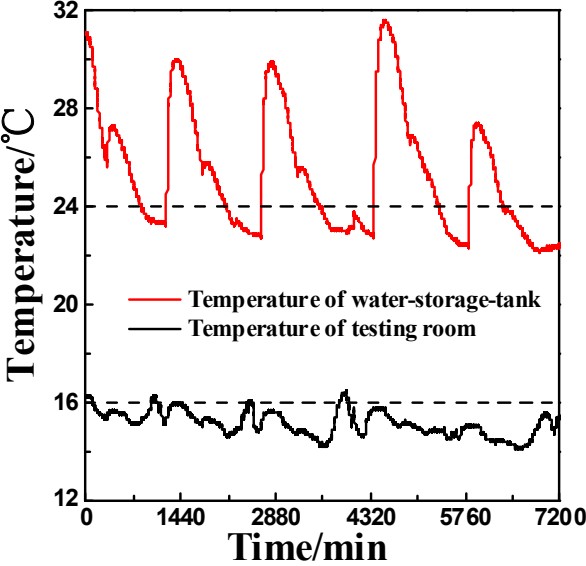

**Figure 5.** Diagram of temperature of the testing room and water-storage-tank over time.

### 3.3. Energy Saving Characteristics

The integrated solar heating system with PCM can meet the heating demand without other heating facilities. It is an economic and environmental protection heating device, which meets the needs and development situation of heating on the Northwest Plateau of China. Evaluating the energy-saving performance of the integrated solar heating system according to the energy-saving rate. Figure 6 is a diagram of the average temperature of the testing room and contrast room. As shown in Figure 6, the red and black columns represent the half-month average temperatures of the testing room with PCM (T1) and the contrast room (T2), respectively. Taking 16 °C as the lowest temperature that the room needs to achieve, the energy that the testing room needs to save, $\Delta T = 16 - T2$, and the energy that the testing room has saved is $ES = T1 - T2$, so that the energy saving rate is $ESR = ES/\Delta T$. Figure 7 is the energy saving rate over time. As shown in Figure 6, the average temperatures of the testing room in the second half of October and the first half of

November are all greater than 16 °C, and the energy saving effect is 100%. From the second half of November to the second half of December, the energy saving effect becomes smaller. The energy saving rate of the second half of December is only 28.6%, which is the lowest. From the second half of December to the first half of April, the energy saving rate becomes higher, and the total energy saving rate is about 45%.

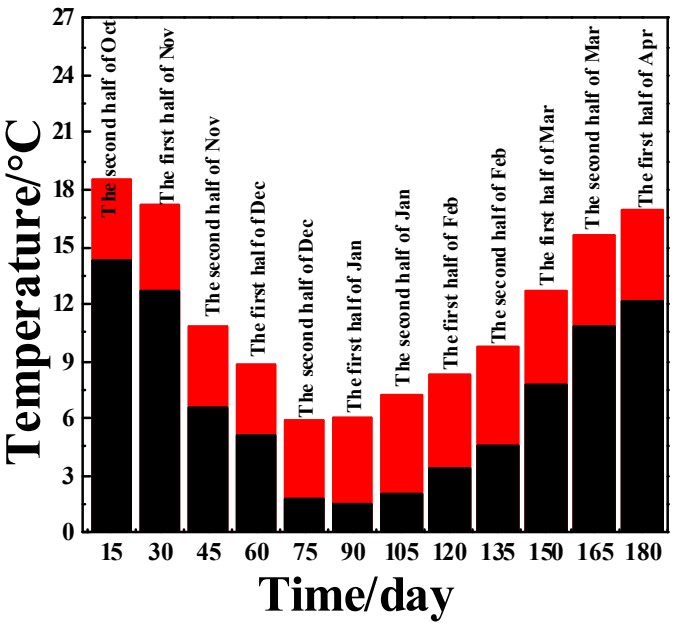

**Figure 6.** Diagram of average temperature of the testing room and contrast room over time.

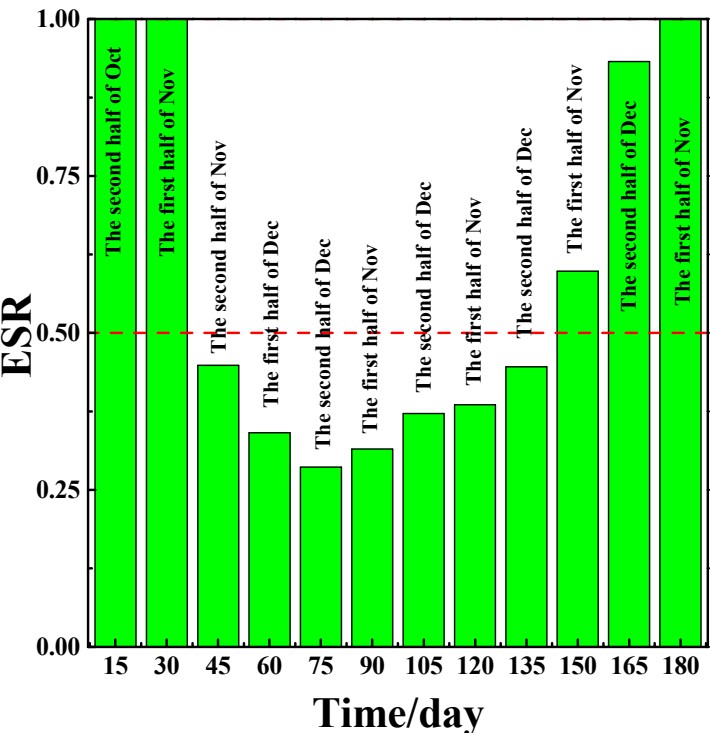

**Figure 7.** Diagram of energy saving rate over time.

## 4. Conclusions

In this study, a new integrated solar heating system coupled with a solar collector, phase change materials and building heating system is studied. The endothermic and exothermic experiment of the phase change material is tested. The stability and comfort

of the integrated solar heating system in a heating period are analyzed. The following conclusions can be drawn from the experimental results: (1) After adding a nucleating agent, the supercooling of the new phase change material decreases, and the supercooling temperature is no more than 2 °C, which has excellent endothermic and exothermic characteristics. (2) The integrated solar heating system coupled with phase change material, a solar collector and building heating system can ensure the stability and continuity of the heating operation. (3) The average temperature of the testing room with PCM is 4.6 °C higher than that of the contrast room. The building with the integrated solar heating system is about 45% more energy efficient than the contrast room. Based on the existing experimental research, this study will further understand and grasp the heat transfer characteristics and thermal storage characteristics of the coupling of a solar water heater, phase change energy storage material and building, and provide the design basis for future engineering applications.

**Author Contributions:** Funding acquisition, H.H.; Investigation, H.Z., O.D. and J.M.; Project administration, H.H.; Writing—original draft, H.H.; Writing—review & editing, H.H. All authors have read and agreed to the published version of the manuscript.

**Funding:** This research was funded by Natural Science Foundation of Hainan Province grant number [221RC586] and Scientific Research Foundation of Hainan Tropical Ocean University grant number [No. RHDRC202102].

**Institutional Review Board Statement:** Not applicable.

**Informed Consent Statement:** Not applicable.

**Data Availability Statement:** Not applicable.

**Acknowledgments:** The authors gratefully thank the Natural Science Foundation of Hainan Province (221RC586) and the Scientific Research Foundation of Hainan Tropical Ocean University (No. RHDRC 202102) for financial support of this work.

**Conflicts of Interest:** The authors declare no conflict of interest.

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
