# Peer review of "Experimental Study on Phase Change Material with Solar Heater System for Building Heating"

_coatings, doi:10.3390/coatings12101476_

Round 1
Reviewer 1 Report
Authors have studied " Experimental Study on Phase Change Material with Solar 2 Heater System for Building Heating" which has good industrial and practical applications. Work on PCM is now the need of the hour. The manuscript is well-written and designed. Recommended for publication with minor revision.
1. Improve the abstract, it should conclude with a line of industrial application at the end of the abstract.
2. Conclusion should be supported with some data.
3. Pls refer to this article (https://doi.org/10.1016/j.ifset.2022.103109) to improve the manuscript.
4. Literature support should be improved.
5. More clarity is required on this work's novelty statement. It should be indicated in the introduction part.
Reviewer 2 Report
English language must be improved to a great extent.
The proposed system and the experimental setup have been tried described in words, but due to the pure quality of English, it is difficult to understand the system and its function. A system sketch with all components used and their connections will increase understanding of the system and its function.
To increase the value of information in Fig. 1 the different rooms (A - E) could have been marked directly in the figure. An additional figure with a floor plan of the entire building with the location of the components, would also increase the understanding of the whole setup. Listing of the rooms in Line 79 is missing room E.
Fig. 3 does not present "outdoor natural environment" as indicated in Lines 158 and 162.
Consider the choice of words in the text in Lines 204-206: ...the energy that testing room needs to saved is ΔT=16-T2, and the energy that testing room have saved is ES=T1-T2, ... . The energy is here given in the unit of temperature. This is not correct. Meanwhile, the energy is proportional to the temperature difference.
Fig. 7 mentioned in Lines 202 and 207 should probably be Fig. 6.
To simplify and increase the understanding of Figs 6 and 7, the horizontal axis can be presented as dates and not in pure digits as it is now. For example 15 Oct / 1 Nov / 15 Nov / 1 Dec etc.
What is the unit for the vertical axis in Fig. 7? Is this ESR? Then it is dimensionless.
Consider switching the order of chapters 3.1 and 3.2. This will give better flow in the use of the figures. That means:
First: 3.1. Comparison of temperature between testing room and contrast room
And then: 3.2. Water temperature in radiator
Round 2
Reviewer 2 Report
No particular comments to the 2nd reviasion.